Effects of short-term nitrogen and phosphorus addition on leaf stoichiometry of a dominant alpine grass

Liu YaLan 1 2 3 4
Liu Bo 5
Yue Zewei 4 6
Zeng Fanjiang 1 2 3 4
Li Xiangyi 1 2 3 4
Li Lei 1 2 3 4 lilei@ms.xjb.ac.cn
1 Xinjiang Key Laboratory of Desert Plant Roots Ecology and Vegetation Restoration, Xinjiang Institute of Ecology and Geography, Chinese Academy of Sciences , Urumqi , China
2 Cele National Station of Observation and Research for Desert-Grassland Ecosystem in Xinjiang , Cele, Xinjiang , China
3 State Key Laboratory of Desert and Oasis Ecology, Chinese Academy of Sciences , Urumqi , China
4 University of Chinese Academy of Sciences , Beijing , China
5 Shandong Provincial Key Lab. of Soil Conservation and Environmental Protection, College of Resources and Environment, Linyi University , Linyi , China
6 Key Laboratory of Ecosystem Network Observation and Modeling, Institute of Geographic Sciences and Natural Resources Research, Chinese Academy of Sciences , Beijing , China
Kang Xiaoming
Electronic publication date: 2021 Dec 22
Publication date: 2021
Volume: 9
Electronic Location ID: e12611
Received 2021 Aug 9; Accepted 2021 Nov 18
Copyright: © 2021 Liu et al.
Copyright year: 2021
Copyright holder: Liu et al.
License: This is an open access article distributed under the terms of the Creative Commons Attribution License, which permits unrestricted use, distribution, reproduction and adaptation in any medium and for any purpose provided that it is properly attributed. For attribution, the original author(s), title, publication source (PeerJ) and either DOI or URL of the article must be cited.
License URL: https://creativecommons.org/licenses/by/4.0/

Keywords: Nutrient addition, Leaf nutrient concentration, Leaf stoichiometry, Alpine grazing grassland

Funding: National Natural Science Foundation of China 41807335 Shandong Provincial Natural Science Foundation ZR2020MC040, ZR2017MC029 Youth Innovation Promotion Association of the Chinese Academy of Sciences 2020434 National Postdoctoral Program for Innovative Talents BX201700279 This research was supported by the National Natural Science Foundation of China (41807335), the Shandong Provincial Natural Science Foundation (ZR2020MC040, ZR2017MC029), the Youth Innovation Promotion Association of the Chinese Academy of Sciences (2020434), and the National Postdoctoral Program for Innovative Talents (BX201700279). The funders had no role in study design, data collection and analysis, decision to publish, or preparation of the manuscript.

==============================
The effects of increasing nitrogen (N) and phosphorus (P) deposition on the nutrient stoichiometry of soil and plant are gaining improving recognition. However, whether and how the responses of N cycle coupled with P of the soil–plant system to external N and P deposition in alpine grassland is still unclear. A short-term external N and P addition experiment was conducted in an alpine grazing grassland in the KunLun Mountain to explore the effects of short-term N and P addition on the nutrient stoichiometry in soil and plant. Different rates of N addition (ranging from 0.5 g N m−2 yr−1 to 24 g N m−2 yr−1) and P addition (ranging from 0.05 g N m−2 yr−1 to 3.2 g P m−2 yr−1) were supplied, and the soil available N, P, leaf N and P stoichiometry of Seriphidium rhodanthum which dominant in the alpine ecosystem were measured. Results showed that N addition increased soil inorganic N, leaf C, leaf N, and leaf N:P ratio but decreased soil available P and leaf C:P. Furthermore, P addition increased soil available P, leaf P, soil inorganic N, leaf N, and leaf C and reduced leaf C:N, C:P, and N:P ratios. Leaf N:P was positively related to N addition gradient. Leaf C:P and leaf N:P were significantly negatively related to P addition gradient. Although external N and P addition changed the value of leaf N:P, the ratio was always lower than 16 in all treatments. The influences of P addition on soil and plant mainly caused the increase in soil available P concentration. In addition, the N and P cycles in the soil–plant system were tightly coupled in P addition but decoupled in N addition condition. The nutrient stoichiometry of soil and leaf responded differently to continuous N and P addition gradients. These data suggested that the alpine grazing grassland was limited by P rather than N due to long-term N deposition and uniform fertilization. Moreover, increasing P addition alleviated P limitation. Therefore, the imbalanced N and P input could change the strategy of nutrient use of the grass and then change the rates of nutrient cycling in the alpine grassland ecosystem in the future.

Introduction

Nitrogen (N) and phosphorus (P) are major limiting elements that determine the plant fixation of carbon (C) (Sardans, Rivas-Ubach & Peñuelas, 2012; Mo et al., 2015; Wang et al., 2020), and the balance of N and P supply is crucial for the terrestrial ecosystems to maintain stability (Li et al., 2020; Yang, 2018). However, previous studies have reported that the amount of N deposition to the ecosystems increased from 32 Tg year−1 to 200 Tg year−1 (Lamarque et al., 2013), while the amount of P increased from 1.7 Tg year−1 to 3 Tg year−1 over the past 100 years (Wang et al., 2014). This imbalance and increasing external N and P supply induced by human activities and global change profoundly exert influences on the nutrient cycle between soil and plant and then alter the structure and functions of an ecosystem (Galloway, Howarth & Michaels, 1996; Bennett, Carpenter & Caraco, 2001; Zhu et al., 2016). Many simulated N and P addition experiments were conducted widely on the global scale to explore how the imbalance of N and P deposition influences the N and P dynamics of the soil–plant system (Lü et al., 2016; Deng et al., 2017). Many studies have reported that terrestrial ecosystems limited by N have been shifted to limited by P or co-limited by N and P based on the analysis of soil–plant nutrient stoichiometry (Tian & Niu, 2015; Peñuelas et al., 2013).

Leaf nutrient concentration was a widely used index of nutrient limitation and the growth of plants (Vitousek 1998; Wright et al., 2005). Leaf nutrient concentration is generally regarded as related to soil nutrient availability (Zhang et al., 2017; Li et al., 2016). Positive relationships between leaf nutrients concentration and soil nutrients availability were also widely observed (Chapin, Vitousek & van Cleve 1986; Chen et al., 2015). How N addition influences N cycle and how P addition influences P cycle homogeneously between soil and plants have been studied widely to predict the influences on the ecosystem of N and P deposition (Wright et al., 2005; Li et al., 2021). For example, Yang (2018) reported that P addition significantly and positively increased leaf and soil P concentrations. Yuan & Chen (2015) reported that N addition improved leaf N and decreased leaf nutrient resorption efficiency by increasing soil N concentration in some N-limited ecosystems. However, studies on how N concentrations in soil and plants respond to P addition and P concentrations respond to N addition are lacking and inconsistent (Zhang et al., 2017; Li et al., 2020). The effects of N addition on leaf P concentration were demonstrated to be neutral (Chen et al., 2015; Tian & Niu, 2015; Peñuelas et al., 2013), positive (Song & Hou, 2020), and negative (Sardans et al., 2015). Deng et al. (2017) found that N addition decreased leaf P concentration in tropical forests in a meta-analysis. Liu et al. (2021) reported that short term P addition alone did not affect soil respiration, but augments the effects of N addition on soil respiration. However, some studies have also shown that N addition exerted no significance on leaf P concentration in tropical forests. The inconsistent results implied that the influences of imbalanced nutrient supply on patterns of coupling relationship between N and P are more complex than expected and they need to be further investigated.

External nutrient supply could reorder plant C, N, P, and other nutrient allocation due to promoting plant growth and other homeostatic abilities (Tian et al., 2018; You et al., 2018). Leaf C:N:P could indicate the growth of plants to a greater extent (Li et al., 2021; Chen et al., 2015). Moreover, leaf N:P ratio was widely used to show and predict ecosystem limitation (Güsewell, 2004; Zhang et al., 2017; Chen et al., 2015), Leaf N:P always increases with N addition (Fujita et al., 2010), implying that N limitation was alleviated and P limitation was aggravated. However, the ratio has some restrictions. Koerselman & Meuleman (1996) reported that N:P ratios of <14 and >16 imply N and P limitation, respectively. Whereas Güsewell (2004) showed that 10 and 20 were the threshold values for N and P limitations in the terrestrial ecosystem from a meta-analysis basis. Moreover, present studies confirmed that the critical values of N:P ratios are determined by species, the ecosystem nutrient limitation type, and other factors (Oheimb et al., 2010; Xu et al., 2014). Therefore, to understand the mechanism of N and P coupled with respond to N and P addition, exploring how the leaf N:P ratio dominating plants in different ecosystems respond to nutrient addition is necessary.

Alpine grassland plays a crucial role in terrestrial ecosystems, which was almost regarded as limited by N or co-limited by N and P (Xu et al., 2014; Li, Niu & Yu, 2016). Some studies reported that climate change and human activity (e.g., fertilization N) brought more uncertainties factors, which lead to N limitation being shifted to P limitation in grassland (Peñuelas et al., 2012; Sardans, Rivas-Ubach & Peñuelas, 2012). However, previous studies about the effect of fertilization on grassland ecosystem limitation mostly focused on the temperate grassland ecosystem, but the alpine grazing grassland has been ignored. These hypotheses have not been tested in alpine grassland (Sardans, Rivas-Ubach & Peñuelas, 2012).

Here, a simulated external nutrient input experiment was conducted in an alpine grassland, which is an important pasture of Kunlun Mountain in Xinjiang Province (Fang et al., 2013). Based on the aforementioned information, we hypothesize (1) leaf and soil nutrient concentration could improve with increasing N and P addition; (2) leaf N and P are linked when responding to N and P addition; (3) the alpine grassland limited by N or co-limited by N and P;. And soil nutrient availability, leaf nutrient concentrations, and leaf stoichiometry after short multi-level N and P addition experiments were explored to quantify these hypothesizes.

Materials and Methods

Study region and experimental design

This study was carried out in an alpine grassland in the Kunlun Mountain, Xinjiang Province (80°35′08″E, 36°08′02″N) in 2017. The altitude of this region is 3,236 m. In 2017 and 2018, the mean annual temperature was 3.2 °C, and it ranges from −7.8 °C in December to 13.2 °C in July. The mean precipitation is 482 mm, mostly occurring from May to September. Long-term meteorological data in this site was deficient. The period of herbed of this site was more than 20 years, and Seriphidium rhodanthum is the main dominant grass in this alpine grassland covers more than 70% of the total aboveground biomass (Li et al., 2020).

The topographic condition in the research site shows uniform. A wide gradient N and P addition experiment was conducted, including the following: CK (control); N1 (0.5 g N m−2 yr−1), N2 (1 g N m−2 yr−1), N3 (2 g N m−2 yr−1), N4 (3 g N m−2 yr−1), N5 (6 g N m−2 yr−1), N6 (12 g N m−2 yr−1) and N7 (24 g N m−2 yr−1) for N addition of urea, and P1 (0.05 g P m−2 yr−1), P2 (0.1 g P m−2 yr−1), P3 (0.2 g P m−2 yr−1), P4 (0.4 g P m−2 yr−1), P5 (0.8 g P m−2 yr−1), P6 (1.6 g P m−2 yr−1), and P7 (3.2 g P m−2 yr−1) for P addition of KH2PO4. The application rates were adopted from the study of Yue et al. (2016), which was conducted in Bayanbulak, Xinjiang. The treatments adopted a completely randomized design, and each has four replicate plots (total of 60 plots). Every plot was 3 m × 2 m and separated by 2 m buffer. In May of 2017, N and P fertilizers were mixed with soil evenly and spread to the plot surface during rainy days.

Plant and soil sampling and measurements

Leaf and soil samples were collected in September 10–12, 2017. Mature leaf samples were collected in the center of the subplot (1 m × 1 m) and dried at 75 °C for 48 h after removing impurities. Dried leaves were ground and sieved by 1 mm mesh for elemental analysis. Leaf C and leaf N were determined by a CN auto-analyzer (Eurovector, Milan, Pavia, Italy). Leaf P was analyzed by Mo-Sb colorimetric method after persulfate oxidation (Sparks, 1996). Leaf C:N, C:P, and N:P ratios were calculated on a mass basis.

Four soil samples were collected from each plot randomly by a 2 cm-diameter soil drilling sampler at 0–10 cm depth and then combined into one sample. All samples were sieved by 2 mm mesh to clear roots and debris after being dried at 105 °C for 48 h for chemical analyses. Soil available N concentration was determined on an auto-analyzer (FIAstar 5000, Foss Tecator, Denmark) after extracting by 2 mol L−1 KCl solution. The soil available P of each treatment was measured via ammonium molybdate method after extraction using 0.5 mol L−1 NaHCO3 solution.

Statistics analysis

One-way ANOVA was used to detect the influences of different N and P additions rates on soil inorganic N; soil available P; leaf C, N, and P concentrations; and leaf C:N, C:P, and N:P ratios. General linear regression models were conducted to measure (1) the relationships between leaf N and leaf P in N addition and P addition treatments; (2) the relationships between leaf C, N, and P concentrations and soil inorganic N and available P concentrations in N and P additions; (3) and the relationships between leaf N:P and soil N:P. Least significant difference method was conducted to test the significant differences among various nutrient addition rates. SPSS 18.0 was used to analyze all statistical data, and the level of P = 0.05 was set to test the significance.

Results

Response of Soil inorganic N and available P concentration to nutrient addition

Soil inorganic N concentration did not significantly respond to N addition (Fig. 1A). N addition decreased soil available P concentration significantly (Fig. 1B), but there was no significant difference between seven N addition rates. This finding indicated that different N addition gradients are not the factor that affect soil inorganic N and available P concentration.

Figure 1 (A–D) Effects of nitrogen (N) and phosphorus (P) addition on soil inorganic N and available P concentrations in the alpine grassland.

CK indicate control treatment. N1, N2, N3, N4, N5, N6 and N7 indicate 0.5, 1, 2, 3, 6, 12 and 24 g N m−2 yr−1. P1, P2, P3, P4, P5, P6, P7 indicate 0.05, 0.1, 0.2, 0.4, 0.8, 1.6, 3.2 g P m−2 yr−1. Different small letter indicates significant difference between treatments (P < 0.05).

P addition increased soil inorganic N (Fig. 1C). And significant differences were noted between different P addition rates. However, only P7 had a significant influence on soil available P concentration, and no obvious difference was observed in P1–P6 (Fig. 1D), suggesting that P addition gradient only influences soil inorganic N and not available P.

Response of Leaf stoichiometry to nutrient addition

N addition increased leaf C and N significantly but exerted no effect on leaf P (Figs. 2A–2C). Leaf C showed no obvious difference between seven N addition rates. P addition significantly affected leaf C and P (Figs. 2D and 2F). And only P6 and P7 increased leaf N significantly. Leaf C and leaf N also showed no obvious difference between different P addition rates.

Figure 2 (A–F) Effects of N and P addition on leaf C, N and P concentrations of Seriphidium rhodanthum.

CK indicates control treatment. N1, N2, N3, N4, N5, N6 and N7 indicate 0.5, 1, 2, 3, 6, 12 and 24 g N m−2 yr−1, respectively. P1, P2, P3, P4, P5, P6, P7 indicate 0.05, 0.1, 0.2, 0.4, 0.8, 1.6, 3.2 g P m−2 yr−1, respectively. Different small letter indicates significant difference between treatments (P < 0.05).

Leaf N:P ratio was positively related to N addition gradient overall, but leaf C:N ratio decreased with N5–N7 addition. Furthermore, N addition did not obviously change leaf C:P ratio (Figs. 3A–3C). P addition decreased leaf C:P and N:P ratios significantly (Figs. 3D, 3F). Obvious negative relationships were found between the two ratios and P addition gradient. Significant differences were also noted among seven P addition rates. However, only P7 decreased leaf C:N ratio; other rates did not have any influence on this ratio. In addition, although N addition increased leaf N:P ratio and P addition decreased it, the ratio was always lower than 16 in all treatments.

Figure 3 (A–F) Effects of N and P addition on leaf C:N, C:P, N:P of Seriphidium rhodanthum.

CK indicates control treatment. N1, N2, N3, N4, N5, N6 and N7 indicate 0.5, 1, 2, 3, 6, 12 and 24g N m−2 yr−1, respectively. P1, P2, P3, P4, P5, P6, P7 indicate 0.05, 0.1, 0.2, 0.4, 0.8, 1.6, 3.2 g P m−2 yr−1, respectively. Different small letter indicates significant difference between treatments (P < 0.05).

Moreover, no relationship was found between leaf N and leaf P under the condition of N addition, but a positive relationship was observed in P addition treatment (Fig. 4).

Figure 4 Relationships between (A) leaf N and (B) leaf P by by general linear regression analysis.

Relationship of nutrient characteristics between soil and plant

In N addition treatments, leaf C had a significant and positive relationship with soil inorganic N and a negative relationship with soil available P (Figs.5A and 5D). Leaf N exhibited negative relationships with soil available P, but it had no relation to inorganic N (Figs. 5B and 5E). Leaf P showed no relationship with either soil inorganic N or available P (Figs. 5C and 5F). In P addition treatments, leaf P had significant and positive relationships with soil inorganic N and available P (Figs. 6C and 6F). And there was no correlation between leaf C, leaf N, and soil nutrient concentration (Figs. 6A, 6B, 6D, and 6E). In addition, leaf N:P ratio was positively correlated with soil N:P in P addition treatments but was not correlated in N addition treatments (Fig. 7).

Figure 5 (A–F) Relationships between leaf C, N, P and soil inorganic N, available P in N addition by general linear regression analysis.

Figure 6 (A–F) Relationships between leaf C, N, P and soil inorganic N, available P in P addition by general linear regression analysis.

Figure 7 Relationships between soil N:P and leaf N:P in (A) N and (B) P addition by general linear regression analysis.

Discussion

Alpine grassland is commonly regarded as being limited by N or co-limited by N and P. The findings in this study exhibited that the alpine ecosystem more limited by P addition than N addition. Furthermore, N and P were tightly linked when responding to P addition but not to N addition.

Effects of nutrient addition on soil and plant nutrient concentration

Consistent with the hypotheses in this study, N addition increased leaf C and leaf N concentrations, also consistent with the previous findings that showed N addition improved N concentrations in leaf and soil and enhanced the ability of C fixation (Deng et al., 2017; Phoenix et al., 2004). However, different to the hypothesis that soil and leaf nutrients will increase with nutrient addition, the results in the present study showed that N addition had a negative effect on soil available P and a neutral effect on soil inorganic N and leaf P. That similar to the study of Yang (2018) which suggested that N addition have no influence on soil N in N-saturated ecosystem. And although N addition promoted plant growth, no nutrient dilution effect was observed on leaf P. The results suggested that grass could adjust the biochemical characteristics to maintain leaf nutrient stability, similar to the study of Phuyal et al. (2008), who demonstrated that plants could not downregulate or even increase P uptake ability to maintain leaf P concentration to respond to increasing biomass with N addition in no N-limited ecosystem. The possible reason for the reduction in soil available P with N addition may be that N addition increased the P requirement of plant with the promotion of plant growth (Lu et al., 2018). In addition, the lower P input in natural conditions, the greater P uptake ability of plants, and the sequestration of litter are the reasons for the significant decrease in soil available P (Deng et al., 2017; Vitousek et al., 2010).

Contrary to the previous studies that showed soil and leaf nutrient concentrations varied with nutrient addition rates (Lü et al., 2013; Yang, 2018), the results of the present study showed that soil and leaf nutrient concentrations had no obvious trend with increasing N addition rates. These neutral effects might caused by the concentration of soil inorganic N in N1 treatment were sufficient to support plant growth, and soil P was the major limiting factor that restricts plant growth rather than N (Deng et al., 2017). The findings were also supported by the results of the relationships between leaf and soil nutrient concentrations through general linear regression analysis. Leaf C and leaf N exhibited a negative and significant relationship with soil available P. Leaf N showed no relationship with soil inorganic N in N addition treatments, thereby confirming that plant growth is mainly limited by soil available P.

P addition increased leaf C, N, and P and soil inorganic N concentrations, similar to previous studies that showed P addition increased the nutrient concentrations of leaf and soil in P-limited ecosystem (Li et al., 2016; Yuan & Chen, 2015). In contrast to the general conclusion that soil P availability could improve with increasing P addition in P-limited ecosystems (Menge & Field, 2007; Xu & Timmer, 1999), the data in the present study showed that soil available P had no significant difference between CK and different P addition rates except P7 treatment. Even though significant differences were found in leaf P among the seven P addition rates, a positive and significant relationship was observed between leaf P and soil available P concentration in P addition treatments. Thus, S. rhodanthum could absorb P from soil indirectly and quickly. The extra P content of P1–P6 treatments was mainly used to support the high P demand induced by the promotion of plant growth and grazing (Medina-Roldán, Paz-Ferreiro & Bardgett, 2012; Phoenix et al., 2004). The addition of P7 was too much for plant growth, and it could conserve P in soil (Deng et al., 2017). Moreover, the results of the experiments in the present study showed that leaf N was not correlated with leaf P when responding to N addition but significantly correlated to leaf P in the P addition treatments. These results were also similar to those of studies that showed plants respond to nutrient addition in N-saturated ecosystems (Yang, 2018; Huang et al., 2016).

Effect of nutrient addition on leaf stoichiometric ratio

In this study, N addition exerted a negative and significant effect on leaf C:N ratio but had no effect on C:P ratio. Leaf C:N and C:P ratios decreased with increasing P addition gradient. These results suggested that S. rhodanthum has dissonant N and P use strategies in different nutrient addition condition. And this finding was similar to the conclusion, which has been accepted widely, that nutrient use strategy varies with different soil nutrient-limitation types (Yuan & Chen, 2009; Sterner & Elser, 2002). The change in leaf C:N and C:P ratios could influence the C:N and C:P ratios of soil through changing the decomposition rates of litter and microbial community composition in general (Güsewell & Gessner, 2009; Chapin, Matson & Mooney, 2002). Zhang, Chen & Ruan (2018) reported that the increasing N concentration in soil and leaf could accelerate litter decomposition and influence the N cycling between soil and plant. Some studies also that reported that N addition could affect soil properties, such as pH, in long-term external experiments and excessive P addition could increase soil pH and then change the strategy of plant nutrient use (Zhang et al., 2008; Hinsinger, 2001; Lu et al., 2018; Yang, 2018). Therefore, the change in leaf C:N:P ratio could have some profound influences on plant nutrient use strategy and then influence this ecosystem with imbalanced N and P input over a long period of time.

The N:P ratio <14, which means the ecosystem is limited by N, and the ratio >16, which means it is limited by P, have been generally shown (Koerselman & Meuleman, 1996). Many studies reported that the threshold of the ratio to indicate ecosystem limitation was not uniform, which depended on plant species and ecosystem types (Güsewell, 2004; Chen et al., 2015). In the present study, the N:P ratio in CK treatment was approximately 12. In N addition treatments, a positive and significant relationship was found between the ratio and N addition rates, and the N:P ratio was always below 16 in the N addition treatments. P addition decreased leaf N:P ratio from 12 in CK treatment to 7 in P7 treatment, and the ratio had a negative and significant relationship with P addition rates.

The results indicated that external P addition not only changed the P concentrations in soil and leaf and the leaf C:P and N:P ratios but also influenced soil and leaf N concentrations and leaf C:N ratios. Leaf P concentration had a positive and significant relationship with leaf N concentration in P addition treatments, suggesting that N and P were tightly coupled in the soil–plant system under P addition treatment in ecosystems (Elser et al., 2000; Ågren, Wetterstedt & Billberger, 2012). However, the results in the present study showed no relationship between leaf N and leaf P in N addition treatments. Leaf N:P ratio was positively related to soil N:P ratio in P addition, but it was not sensitive to the ratio in N addition treatments. Furthermore, the extent of change of N:P in P addition was greater than in N addition. These results suggested that S. rhodanthum is influenced by P addition to a greater extent, similar to the studies that leaf nutrient concentration and stoichiometry were more sensitive to P addition rather than to N addition in N-saturated ecosystems (Deng et al., 2017; Yang, 2018; Chen et al., 2015; Yuan & Chen, 2015). Therefore, these results highlighted that P rather than N was the major limiting factor in this ecosystem. The plausible reason was that the grassland was limited by P, different from previous studies, which showed that the grassland was mostly limited by N, possibly because long-term imbalanced fertilization increased the soil inorganic N concentration and then changed the limitation from N to P (Lebauer & Treseder, 2008; Li et al., 2016). In addition, the threshold of N and P limitation was approximately 12 rather than 14 and 16. Thus, the conclusion of Koerselman & Meuleman (1996) could not be adopted to this alpine grazing grassland, and the understanding of the mechanism of leaf N:P ratio response to external nutrient addition in different ecosystems needs further investigation.

Conclusion

In conclusion, our results were different to our hypothesizes: (1) P addition not only increased P concentration but also changed N concentrations in soil and leaf. N addition increased soil inorganic N and leaf but decreased soil available P and had no effect on leaf P. External nutrient addition change C:N, C:P, and N:P ratios by altering nutrient concentration in leaf. The varied leaf nutrient characteristics showed that external nutrient addition changes the strategy of nutrient use and would change the rates of nutrient cycling between plant and soil; (2) N and P cycling in the soil-plant system were link tightly when respond to external P addition but independent in N addition and the plant was more sensitive to P addition rather than N addition; (3) this ecosystem is limited by P rather than N due to long-term N deposition and imbalanced fertilization supply. P addition alleviated P limitation. In addition, the study we conducted was a short-term and N and P alone fertilization experiment, but previous findings showed that the responsive of nutrient addition on the plant was depend on the period of experiment time and there were some differences between N and P addition alone and combined (Clark & Tilman, 2008; Li et al., 2020; Peng et al., 2019). Therefore, further studies were needed to improve understanding of the effects of long-term and NP combined addition on the nutrient cycle in the soil-plant system.

Supplemental Information

Supplemental Information 1 Raw data.

Soil NO3−-N, NH4+-N, inorganic N, available P in seven N and P addition gradient

Click here for additional data file.

Supplemental Information 2 Leaf nutrients in nutrient addition.

Leaf C, N, P in seven N and P addition gradient

Click here for additional data file.

Supplemental Information 3 Leaf C:N, C:P, N:P.

Click here for additional data file.

Additional Information and Declarations

Competing Interests

Author Contributions

Data Availability

The authors declare that they have no competing interests.

YaLan Liu conceived and designed the experiments, performed the experiments, analyzed the data, prepared figures and/or tables, and approved the final draft.

Bo Liu conceived and designed the experiments, performed the experiments, authored or reviewed drafts of the paper, and approved the final draft.

Zewei Yue conceived and designed the experiments, performed the experiments, authored or reviewed drafts of the paper, and approved the final draft.

Fanjiang Zeng performed the experiments, authored or reviewed drafts of the paper, and approved the final draft.

Xiangyi Li performed the experiments, authored or reviewed drafts of the paper, and approved the final draft.

Lei Li conceived and designed the experiments, performed the experiments, authored or reviewed drafts of the paper, and approved the final draft.

The following information was supplied regarding data availability:

Raw data are available in the Supplemental Files.

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
