# Peer review of "Effects of short-term nitrogen and phosphorus addition on leaf stoichiometry of a dominant alpine grass"

_PeerJ, doi:10.7717/peerj.12611_

## Round 0.1 · original submission · Major Revisions

Dear Dr. Liu and co-authors,

I just received the reviews of your manuscript. Please, consider all comments and suggestions provided by both reviewers during the revision of your manuscript. A more detailed description of the method and results section is also needed before the acceptance.

A comprehensive revision of the English of the manuscript is necessary before submitting the new version.

Don't forget to include a letter response along with the revised version of the manuscript. In this letter you must respond point by point to each question.

Best regards,

Xiaoming Kang

Reviewer 1 ·

Basic reporting

Clear and unambiguous, professional English used throughout
Comments: English writing is not authentic. Too many long sentences make it difficult for readers to read this article. Some sentences are difficult to follow. For example, Line 12-14 “The effects of increasing N and P deposition on nutrient concentration and stoichiometry of soil and plant are gaining improving recognition, but it is unclear whether and how the mechanism of N cycle coupled with P of the soil-plant system response to increasing imbalance external N and P supply in alpine grassland” would be better to understand if change it to “The effects of increasing N and P deposition on nutrient concentration and stoichiometry of soil and plant are gaining improving recognition. However, whether and how the responses of N cycle coupled with P of the soil-plant system to external N and P deposition in alpine grassland is still unclear”. I suggest the author find a native English speaker or an ecological expert who is good at English articles writing to further polish the English language.

Literature references, sufficient field background/context provided
Comments: No comments

Professional article structure, figures, tables. Raw data shared
Comments: Results are too brief. The results should first show whether the addition of nitrogen or phosphorus significantly affects the relevant variables (show F and P value), and then at least explain which level treatment significantly improves or reduces the variable and the extent of change compared with the control. For example,
I have found an interesting pattern that N1, N3, and N7 significantly elevated NO3--N, but other levels did not affect it. Why? I suggest authors explain it.

Self-contained with relevant results to hypotheses
Comments: No comments

Experimental design

Original primary research within Aims and Scope of the journal.
Comment: Yes
Research question well defined, relevant & meaningful. It is stated how research fills an identified knowledge gap.
Comment: Yes
Rigorous investigation performed to a high technical & ethical standard.
Comment: Yes
Methods described with sufficient detail & information to replicate.
The description of the method section is too brief. Randomized block design or completely randomized design? What was the way of fertilization? What was the fertilization time? What were the rules for leaf selection? How many plant leaves are collected in each sample plot or how many weight leaves are collected? How is the soil preserved? What is the concentration of soil KCl extract? How long will it take? The above questions need to be described in detail or given references. In addition, authors conducted regression in the last 2 figures, but only describe correlation analysis in the method.

Validity of the findings

Impact and novelty not assessed. Meaningful replication encouraged where rationale & benefit to literature is clearly stated
Comment: no comments

All underlying data have been provided; they are robust statistically sound & controlled.
Comment: yes

Conclusions are well stated, linked to original research question & limited to supporting results
Comment: yes

Additional comments

1.The author used “leaf nutrient concentration” and “stoichiometry” at the same time in the MS. These two words have the same meaning, I suggest change ““leaf nutrient concentration” to “stoichiometry”.
2. The numbers in many units in the text should be superscript, such as line 17, line 93-96, please check the whole MS, and revise it.
3. There should be a space between the number and the unit, such as line 97-98. please check the whole MS, and revise it.
4. Fig. 1 The data of ammonium nitrogen and nitrate nitrogen completely overlap and are difficult to distinguish. In addition, you say “Different small letter indicates significant difference between treatments” in the title, but where are the letters?
5. Error bars in the figures are SD or SE, clarify.
6. R2 value should be displayed in Fig 3A
7. change “correlation” to “regression” in Fig. 4 and Fig. 5

Reviewer 2 ·

Basic reporting

no comment

Experimental design

no comment

Validity of the findings

no comment

Additional comments

Common comment
The authors reported the effects of short-term nutrient addition on leaf nutrient concentration and stoichiometry of a dominant alpine grass. I think the research is interesting and valuable. The manuscript is well organized and has provided sound conclusion. I cannot find many errors in the manuscript. The language is ok, but it would be easier to read if the authors double-check the grammar and improve the expression. However, there are still some small errors need to be addressed. Thus, in the present form of the manuscript, I suggest a minor revision should be made before the publication in the journal PeerJ.

Specific comment

Abstract
1. I suggest the authors add crucial and detailed stoichiometry data in the abstract.
2. “Nutrient addition” is far too ambiguous and I suggest the author change the expression with detailed treatments.

Introduction
1. Please provide your scientific hypothesis in the introduction section, which is essential to fully report your work.
Materials and methods
1. Please provide detailed collection procedure of the plant and soil samples. What kind of leaf was collected? Young or old leaf?
2. Please specify what tests were done for the general linear regression models and ANOVA.

Results
1. The authors should show the detailed data for all the results.
2. I suggest the authors expand the results section. The current results were far too simple and showed very limited results.
3. No results for basic properties of plant and soil samples? I understand the authors may focus on C:N:P stochiometry, however, basic properties are useful and even necessary.

Language
1. The language of the manuscript is not too bad, but may be better if the authors ask an English-speaker to polish the language.
2. Please check the grammar throughout the manuscript.

Tables and figures
1. Please improve the quality of all figures, which are difficult to read.
2. Use typical expression of numbers instead of “-.523”.

---

## Round 0.2 · accepted · Accept

Dear authors,

I am pleased to inform you that, following the revision made based on the reviewer’s comments, your manuscript is now acceptable for publication in PeerJ.

Best regards

Xiaoming Kang

Reviewer 1 ·

Basic reporting

The author respond all my concerns in detail and made carefully revisions according to my comments. The manuscript has been greatly improved. I don't find any other problems with this paper, so I suggest to accept this paper

Experimental design

No comments

Validity of the findings

No comments

Additional comments

No comments

Reviewer 2 ·

Basic reporting

no comment.

Experimental design

no comment.

Validity of the findings

no comment.